# Short-Term Effects of "Polish Smog" on Cardiovascular Mortality in the Green Lungs of Poland: A Case-Crossover Study with 4,500,000 Person-Years (PL-PARTICLES Study)

Łukasz Kuźma [1] , Anna Kurasz [1,*] , Emil Julian Dąbrowski [1] , Sławomir Dobrzycki [1]
and Hanna Bachórzewska-Gajewska [1,2]

1   Department of Invasive Cardiology, Medical University of Bialystok, 24A Sklodowskiej-Curie St,
    15-276 Bialystok, Poland; kuzma.lukasz@gmail.com (Ł.K.); e.j.dabrowski@gmail.com (E.J.D.);
    slawek_dobrzycki@yahoo.com (S.D.); hgajewska@op.pl (H.B.-G.)
2   Department of Clinical Medicine, Medical University of Bialystok, 24A Sklodowskiej-Curie St,
    15-276 Bialystok, Poland
*   Correspondence: annaxkurasz@gmail.com; Tel.: +48-857468496

**Abstract:** Previous studies conducted in highly polluted areas have reported associations between air pollution and daily mortality. The Green Lungs of Poland are characterized by unique natural features and a moderate pollution level. We aimed to assess the short-term impact of air pollution on cardiovascular (CVD)-, acute coronary syndrome (ACS)-, and cerebrovascular-related (CbVD) mortality. An analysis with 4,500,000 person-years and a time-stratified case-crossover design was performed. The interquartile range increase in the $PM_{2.5}$ (OR 1.036, 95% CI 1.016–1.056, $p < 0.001$) and $PM_{10}$ concentration (OR 1.034, 95% CI 1.015–1.053, $p < 0.001$) was associated with increased CVD mortality on lag 0, and this effect persisted on the following days. The effects of PMs were expressed more in association with ACS-related mortality ($PM_{2.5}$-OR = 1.045, 95% CI 1.012–1.080, $p = 0.01$; $PM_{10}$-OR = 1.044, 95% CI 1.010–1.078, $p = 0.01$) and CbVD mortality ($PM_{10}$-OR = 1.099, 95% CI 1.019–1.343, $p = 0.02$). We also noted a higher CVD mortality OR in the cold season for $PM_{10}$ in cities with area-source domination: Białystok ($p = 0.001$) and Suwałki ($p = 0.047$). The short-term impact of PMs on cardiovascular mortality is also observed in moderately polluted areas. This adverse health effect was more apparent in CbVD- and ACS-related mortality, and in the cold season. Further research focusing on the adverse health effects of "Polish smog" is sorely needed.

**Keywords:** air pollution; Polish smog; cardiovascular mortality; nitrogen dioxide; mortality; particulate matter

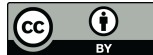

## 1. Introduction

Air quality and its impact on health have been in the spotlight since the infamous Great Smog of London. Since then, it has become clear that the adverse effects are serious and can no longer be ignored. According to the World Health Organization (WHO) estimations, in the last year, ambient air pollution caused 4,200,000 deaths worldwide. These statistics included a total of 29,165 deaths that were attributed to air pollution in Poland [1].

Our study is focused on the Green Lungs of Poland, located in the north-eastern part of the country: an area characterized by unique natural features, large areas covered by forests, a lack of factories, and relatively low industrialization. This region is widely known as the one with the lowest per capita income in the European Union. As it is characterized by low socioeconomic status, in the cold season the residents' suboptimal heating choices pose a major anthropogenic threat to the air quality in the form of area source emissions.

Both the poor heating choices and the specific geographic location of the region, especially at times of frosty Russian Federation weather conditions characterized by high pressure, cold air and sunshine, favor the formation of the phenomenon known as "Polish

smog". The air pollution, rich in compounds such as $PM_{2.5}$, $PM_{10}$, and polycyclic aromatic hydrocarbons (benzo(a)pyrene) from low emissions associated with household heating with solid fuels (coal, wood, and often also waste), imposes detrimental effects on the health and life of the population, in particular in the context of cardiovascular effects [2].

Regardless of the source, the mechanisms linking air pollution with cardiovascular diseases (CVD) remain unclear. Three main theories shed some light on the ways in which inhaled pollutants could affect the cardiovascular system (CVS). The oldest one states that pollutants activate inflammatory cells in the lungs, leading to the release of mediators that alter cardiovascular function. The second one explains its influence via an imbalance in the autonomic nervous system and neuroendocrine pathway. Pollutants stimulate alveolar receptors that activate autonomic reflex arcs influencing cardiovascular homeostasis. The last theory claims that pollutants directly influence CVS. Due to the large alveolar surface, air pollutants translocate via the pulmonary epithelium and enter the blood circulation. It seems like all of the mechanisms overlap, but they lead to one major result: oxidative stress [3–7].

Although the early steps of inhaled pollutants' influence on CVS are still a subject of debates, oxidative stress is widely acknowledged as the leading pathophysiological factor for vascular dysfunction. Reactive oxygen species (ROS) cause vasoconstriction, impair endogenous fibrinolysis, and promote the aggregation of platelets. Chronic endothelial cell injury leads to atherosclerosis, and acute ruptures of plaques cause myocardial infarction or stroke. The high harmfulness of poor air quality may last for days after the exposition, and could be boosted even more by changes in the weather [8–10].

The vast majority of the studies on air pollution were conducted in highly polluted areas, in which patients are exposed to extreme concentrations of pollutants. Taking into consideration the low number of surveys from areas with a moderate level of pollution, we analyzed the relationship between air pollution and cardiovascular mortality in north-eastern Poland [11–13]. The short-term impact of air pollution on cardiovascular (CVD)-, acute coronary syndromes (ACS)-, and cerebrovascular-related (CbVD) mortality was assessed. The examination of the impact of so-called "Polish smog" on mortality in the study area is not without significance. It is the first major study analyzing the impact of this type of pollution on human health covering the data from the past ten years.

## 2. Materials and Methods

### 2.1. Studied Region

The studied region, Podlaskie Voivodeship, is located in north-eastern Poland. With four national parks, large areas covered by forests, a lack of factories, and relatively low industrialization, the region is widely known as the Green Lungs of Poland. The biggest cities of the region are Białystok, Łomża, and Suwałki. Despite their location, the characteristics of the cities contribute to a moderately increased level of air pollution. Białystok, the capital of Podlaskie Voivodeship, is the biggest city, with area sources contributing to the pollution by almost 50%. In Suwałki, almost 85% of the air pollution is emitted by area sources. Łomża plays an important part in transit traffic from Northern and Eastern Europe to Central Europe. Therefore, with almost 60% of the share, linear sources were the dominant sources of $PM_{10}$ in the city's exceedance areas during the analyzed period of time [14–16]. A detailed description of the studied cities is presented in Figure 1.

### 2.2. Mortality Data

Data on mortality were collected from the National Statistical Office in Poland. The records include the information on all of the deaths, along with the age, sex and causes of death, recorded in the Podlaskie Voivodship in the years 2008–2017. In the analysis, we used data from patients registered and residing in the city of Białystok (id commune 206101), Łomża (id commune 206301), and Suwałki (id commune 200702).

According to the codes in the International Classification of Diseases (ICD)—10th Revision, we extracted the data for CVD-related mortality (ICD-10 from I.00 to I.99 for car-

diovascular diseases), ACS-related mortality (ICD-10 codes: I20 for unstable angina, I21.XX for acute coronary syndromes, and I25.XX for ischemic heart disease), and cerebrovascular mortality (ICD-10 codes: I63.XX for cerebral infarction) [17]. Consequently, the mortality data for these codes from the 2008–2017 period were included in the analysis.

Crude death rates (CDR) were calculated as a simple ratio: the number of registered deaths/the mid-year population (per 100,000). To calculate the standardized death rate (SDR), we used a standard European population structure [18]. SDR is calculated as a weighted average of the age-specific death rates of a given population; the weights are the age distribution of that population.

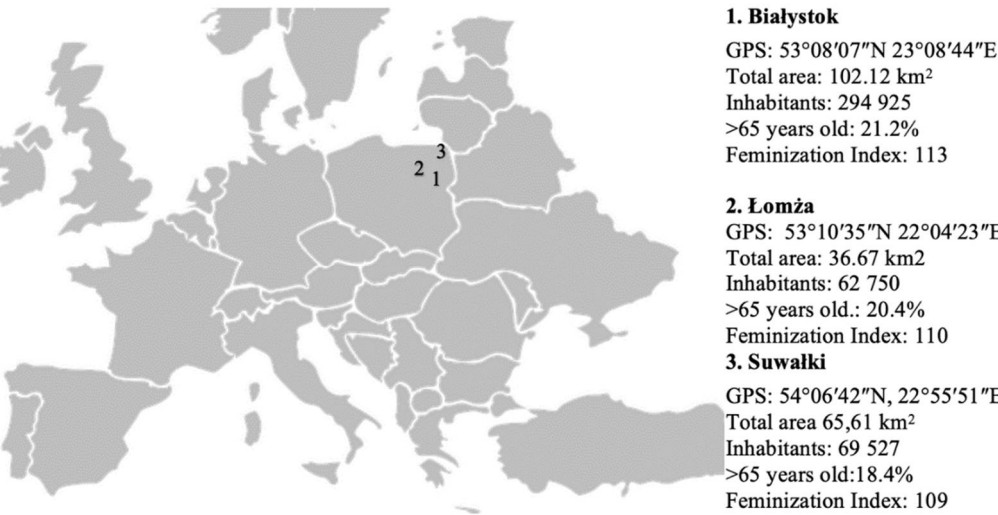

**1. Białystok**
GPS: 53°08′07″N 23°08′44″E
Total area: 102.12 km²
Inhabitants: 294 925
>65 years old: 21.2%
Feminization Index: 113

**2. Łomża**
GPS: 53°10′35″N 22°04′23″E
Total area: 36.67 km2
Inhabitants: 62 750
>65 years old.: 20.4%
Feminization Index: 110

**3. Suwałki**
GPS: 54°06′42″N, 22°55′51″E
Total area 65,61 km²
Inhabitants: 69 527
>65 years old:18.4%
Feminization Index: 109

**Figure 1.** Characteristics of the studies cites.

*2.3. Pollution and Meteorological Data*

The data on air pollution and gases were obtained from the Voivodeship Inspectorate for Environmental Protection. In the analysis, we used the concentration of sulfur dioxide ($SO_2$), nitrogen dioxide ($NO_2$), and particulate matter with diameters of 2.5 μm or less ($PM_{2.5}$) and 10 μm or less ($PM_{10}$).

In Białystok city, the data on the concentrations of $NO_2$, $SO_2$ (except 2013), $PM_{2.5}$ and $PM_{10}$ were obtained from stations located in the center of the city (international code (ID): PL0148A, PL0496A, PL0496A and PL0147). The $SO_2$ concentrations for 2013 were obtained from the suburban station ID PL0149A.

In Łomża city, all of the measurements ($SO_2$, $NO_2$, $PM_{2.5}$, $PM_{10}$) were obtained from a station located in the north part of the city: PL0151A. In Suwałki, only $PM_{2.5}$ and $PM_{10}$ were collected in the analyzed period. All of the measurements were obtained from station PL0152A.

The daily meteorological data, including the mean temperature, the daily level of relative humidity, and the mean atmospheric pressure, were obtained from the Institute of Meteorology and Water Management. In Białystok, we used the data from station Ciołkowskiego Street ID 353230295. In Suwałki we used the data from station ID 354220195. In Łomża, for years 2008–2011, it was the ID 253210210 station; for 2011–2015 it was ID 253220280, and station ID 253220330 for 2016 and 2017.

According to the Voivodeship Inspectorate for Environmental Protection and the Institute of Meteorology and Water Management, the measurements that were used in the analyses are representative for the studied cities.

In the analysis, we used the 24h (midnight to midnight) concentration of air pollutants. The study material lacked about 3.5% of the daily data of the air pollution concertation—days without data were excluded from the analysis.

### 2.4. Study Design and Statistical Analysis

The case-crossover design is a recognized method to assess episodic events following short-term exposure to air pollution. In order to assess the effect of particulate matter and the concentration of gases on mortality we used a time-stratified design. The day of death was defined as the case period, while the control periods included all of the days that were from the same day of the week in the same month as the case period. This method was used to adjust for the effects of seasonality, long-term trends, and the day of the week, giving 3–4 days for each case [19,20]. Thus, the cases serve as their own controls, providing implicit control of all of the known and unknown confounders that are unlikely to vary non-randomly during the time.

In the analysis, the residence was used to link with the exposure to air pollution. We used the weather conditions as a covariate in all of the models. Because of high collinearity, and to minimize that effect, each air pollutant was modeled individually. In the case of noting the influence of more than one pollutant, we additionally created the multi-pollutant model.

We estimated the effects of air pollutants at a single-day lag (from lag 0 to lag 3) and at a multi-day lag (lag 0–1 and lag 0–3), which defines the short-term period as a maximum of 3 days. For example, lag 0 and lag 0–1 correspond to the concentration of pollutants on the death day and moving average for the death day and one previous day, respectively. We conducted separate models for the warm and cold seasons for all of the lags. We defined the seasons as follows: the months from March to the end of September were the warm season, and from October to the end of February were the cold season.

Finally, we created 99 models (due to lack of measurements of $NO_2$ as well as $SO_2$ for Łomża). In the second stage, in order to estimate the pooled effect size of air pollution, the meta-analysis was performed. We created the tri-city model for PMs, and a two-city model for gases' associations with cardiovascular, coronary artery, and cerebrovascular mortality.

The distribution of the variables was evaluated using the Kolmogorov–Smirnov test. Normally distributed variables were expressed as mean values with a standard deviation (SD), and the variables which did not follow a normal distribution were expressed as medians, 1st and 3rd quartiles, and interquartile ranges (IQR). Due to non-normal distribution, a Kruskal–Wallis test was used for the comparative analysis between the mortality differences (except age, for which an analysis of variance was used).

Spearman's rank correlation test was applied for the evaluation of the relationships between the levels of air pollutants.

The association between air pollution and the occurrence of deaths was estimated using an odds ratios with 95% confidence intervals, using conditional logistic regression (CLR). The mean daily values of the air pollution, weather conditions (midnight to midnight) at day of death, or the previous days in the lag and multi-lag structure, were used for the statistical analysis. Days with missing data were excluded from the analysis. In order to avoid collinearity, pollutants were included in the models separately. We used the mean daily temperature, mean daily humidity, and mean daily sea-level atmospheric pressure conditions of the same lag structures with the pollutants (due to the lack of data in Łomża, we used only the temperature and humidity) as covariates in the CLR model. The check for the best transformation of the temperature, humidity and pressure analysis was conducted using various degrees of freedom (df). The final composition of the function was a natural cubic spline of the temperature, humidity, and atmospheric pressure, with 4 df and a natural cubic spline with 3 degrees of freedom for the lag structure. The results are reported as the OR associated with an increase in the interquartile range (IQR). ORs were given for $NO_2$, $SO_2$, $PM_{2.5}$, and $PM_{10}$. The differences in the odds ratio between the warm and cold season were calculated with an increase in 10 $\mu g/m^3$ of the PMs and 1 $\mu g/m^3$ of the gases according to the Altman method [21].

A meta-analysis was performed to estimate the pooled effect size of air pollution. It is a widely acknowledged method used in many multi-center air pollution studies [22,23].

A fixed-effects model was used. The results are reported as the OR associated with an increase in the interquartile range.

The threshold of statistical significance for all of the tests was set at $p < 0.05$.

All of the analyses were performed using MS Excel (Microsoft, 2020, version 16.40, Redmond, WA, USA), XL Stat (Addinsoft, 2020, version 2020.03.01, New York, NY, USA), and Comprehensive Meta-Analysis (version 3, Biostat Inc, 2020, Englewood, NJ, USA). The figures were created using GraphPad Prism (version 9, 2020, San Diego, CA, USA).

The study protocol was approved by the ethics committee of the Medical University of Białystok (R-1-002/18/2019), and was registered at ClinicalTrials.gov (NCT04541498).

## 3. Results

The analyzed region was inhabited by almost 500,000 residents. From 2008 to 2017 in Białystok, Łomża, and Suwałki, we recorded 49,573 deaths—34,005, 8082, and 7486, respectively (Figure 1). Cardiovascular SDR was the highest in Łomża, at 871.42 per 100,000 population/year, and lowest in Białystok, at 637.59 per 100,000 population/year ($p < 0.001$). There were no differences in the age and gender balance between the cardiovascular deaths in the recorded cities (Table 1), (Table S1).

**Table 1.** Mortality in the studied region in the years 2008–2017.

| | Białystok | Łomża | Suwałki | *p* |
|---|---|---|---|---|
| Total deaths, N | 34005 | 8082 | 7486 | |
| Male, % (N) | 52.49 (17851) | 54.54 (4408) | 54.12 (4055) | <0.001 |
| Mean age (SD) | 71.99 (16.59) | 72.66 (15.68) | 71.68 (16.56) | 0.005 |
| CDR [3], (100,000 population/year) | 1153.01 | 1288.13 | 1079.51 | <0.001 |
| SDR [5], (100,000 population/year) | 1480.95 | 1944.57 | 1638.06 | <0.001 |
| Cardiovascular mortality rate, %, (N) | 40.73 (13851) | 39.94 (3228) | 36.36 (2723) | <0.001 |
| Male, % (N) | 46.99 (6508) | 46.97 (1563) | 46.55 (1268) | 0.915 |
| Mean age (SD) | 77.70 (11.88) | 78.19 (11.28) | 76.40 (12.37) | 0.125 |
| CVD [4] CDR, (100,000 population/year) | 469.65 | 530.36 | 392.72 | <0.001 |
| CVD SDR, (100,000 population/year) | 637.59 | 871.42 | 644.84 | <0.001 |
| Acute coronary syndromes rate, %, (N) | 14.11 (4799) | 13.24 (1070) | 11.98 (897) | <0.001 |
| Male, % (N) | 51.42 (2467) | 53.93 (577) | 53.40 (479) | 0.230 |
| Mean age (SD) | 77.45 (10.18) | 77.11 (11.51) | 75.93 (11.76) | 0.013 |
| ACS [1] CDR, (100,000 population/year) | 162.72 | 270.25 | 129.51 | <0.001 |
| ACS SDR, (100,000 population/year) | 217.95 | 170.52 | 207.18 | <0.001 |
| Cerebrovascular mortality rate, %, (N) | 8.02 (2727) | 9.21 (744) | 5.77 (432) | <0.001 |
| Male, % (N) | 40.7 (1110) | 40.1 (298) | 37.0 (160) | <0.001 |
| Mean age (SD) | 79.37 (13.3) | 79.61 (9.4) | 79.18 (10.3) | <0.001 |
| CbVD [2] CDR, (100,000 population/year) | 92.46 | 118.73 | 62.16 | <0.001 |
| CbVD SDR, (100,000 population/year) | 126.84 | 199.75 | 105.94 | <0.001 |

[1] ACS, acute coronary syndromes; [2] CbVD, cerebrovascular disease; [3] CDR, crude death rate; [4] CVD, cardiovascular disease; [5] SDR, standardized death rate.

The distributions of the air pollution concentrations and weather conditions are listed in Table 2 and Figures S1–S3. The median daily concentrations of $PM_{10}$ (23.8 µg/m$^3$, IQR = 16.9), $NO_2$ (13.4 µg/m$^3$, IQR = 9.5), and $SO_2$ (5.8 µg/m$^3$, IQR = 5.8)—except $PM_{2.5}$ (13.3 µg/m$^3$, IQR = 9.1)—were the highest in Łomża. The median daily concentration of $PM_{2.5}$ was the highest in Białystok (16.2 µg/m$^3$, IQR = 15.9), and the median concentrations of $PM_{10}$, $NO_2$, $SO_2$, and $O_3$, were 23 µg/m$^3$ (IQR = 17.2), 13.1 µg/m$^3$ (IQR = 7.8), 2.7 µg/m$^3$ (IQR = 2.8), respectively. In the case of Suwałki, the daily median $PM_{2.5}$ concentration was 9.8 µg/m$^3$ (IQR = 8.7), and of $PM_{10}$ was 18.0 µg/m$^3$ (IQR = 14.3) (Table 2).

The detailed results of the impact of air pollution on mortality in the cities included in the meta-analysis are shown in Tables S2–S4. The highest OR for the IQR (8.7) increase in the $PM_{2.5}$ concentration (OR = 1.10, 95% CI 1.01–1.2, $p = 0.03$) was noted in Suwałki, in comparison with Białystok (OR = 1.06, 95% CI 1.01–1.11, $p = 0.01$; IQR = 15.9) and Łomża (OR = 1.02, 95% CI 0.97–1.08, $p = 0.37$; IQR = 9.1). A similar relation was noted for $PM_{10}$. In Suwałki, the OR for the IQR (14.3) increase was 1.05 (95% CI 1.00–1.11, $p = 0.053$), in Białystok it was OR = 1.03 (95% CI 1.01–1.05, $p = 0.01$; IQR = 17.2), and in Łomża it was

OR = 1.03 (95% CI 0.98–1.08, *p* = 0.23; IQR = 16.9). The effect was noted regardless of the cause of death for lag 0, 1, 2 and multileg lag 0–1 and 0–3. We also noted a higher CVD mortality OR for the cold season than the warm season for $PM_{10}$ in Suwałki at lag 0–3 (*p* = 0.047), and in Białystok at lag 0 (*p* = 0.001) [Table S2]. The opposite relation was noted in Łomża, where the impact of $PM_{10}$ (*p* = 0.03) on the ACS-related mortality was higher in the warm season.

**Table 2.** Statistics for the 24-h (midnight to midnight) concentrations of air pollutants and weather conditions in the studied region.

| Białystok | $PM_{2.5}$ [7] μg/m³ | $PM_{10}$ [8] μg/m³ | $NO_2$ [5] μg/m³ | $SO_2$ [9] μg/m³ | Temp.[6] °C | Hum.[2] (%) | Atm.[1] (hPa.) |
|---|---|---|---|---|---|---|---|
| Median | 16.2 | 23.0 | 13.1 | 2.7 | 7.8 | 83.1 | 1015.5 |
| IQR [3] | 15.9 | 17.2 | 7.8 | 2.8 | 13.3 | 17.0 | 10.7 |
| (1Q–3Q) | 10.9–26.8 | 15.8–33.0 | 9.9–17.7 | 1.5–4.2 | 1.5–7.7 | 73.6–81.4 | 1010–1021 |
| Median—cold season | 25.5 | 28.0 | 14.7 | 3.8 | 1.6 | 89.0 | 1016.5 |
| IQR—cold season | 19.8 16.8–36.6 | 21.5 18.6–40.0 | 8.8 10.5–19.3 | 3.4 2.3–5.6 | 6.8 −1.9–4.9 | 11.9 82.1–94.0 | 14.6 1009.3–1024 |
| Median—warm season | 11.9 | 19.4 | 12.1 | 1.9 | 14.8 | 75.6 | 1015.1 |
| IQR—warm season | 7.3 8.6–15.9 | 12.2 14.1–26.3 | 8.7 9.4–15.7 | 1.8 1.1–2.9 | 6.2 11.5–17.7 | 15.3 68.8–84.1 | 8.0 1011–1019 |
| Łomża | $PM_{2.5}$ μg/m³ | $PM_{10}$ μg/m³ | $NO_2$ μg/m³ | $SO_2$ μg/m³ | Temp. °C | Hum. (%) | Atm. (hPa.) |
| Median | 13.3 | 23.8 | 13.4 | 5.8 | 7.7 | 83.3 | n.d. [4] |
| IQR | 9.1 9.6–18.7 | 16.9 17.1–34.0 | 9.5 9.0–18.5 | 5.8 3.5–9.4 | 13.7 1.6–7.7 | 18.0 73.0–83.3 | n.d. |
| Median—cold season | 22.4 | 32.7 | 15.2 | 9.1 | 1.8 | 89.8 | n.d. |
| IQR—cold season | 16.6 15.4–32.0 | 23.8 22.3–46.1- | 10.4 10.5–20.9 | 6.4 6.2–12.6 | 6.8 −1.8–5.0 | 11.0 83.5–94.5 | n.d. |
| Median—warm season | 12.2 | 20.9 | 11.8 | 4.0 | 15.2 | 75.3 | n.d. |
| IQR—warm season | 8.2 9.1–16.2 | 11.8 15.2–27.1 | 8.0 8.0–16.0 | 3.0 2.5–5.5 | 6.6 11.6–18.2 | 16.0 67.0–83.0 | n.d. |
| Suwałki | $PM_{2.5}$ μg/m³ | $PM_{10}$ μg/m³ | $NO_2$ μg/m³ | $SO_2$ μg/m³ | Temp. °C | Hum. (%) | Atm. (hPa.) |
| Median | 9.8 | 18.0 | n.d. | n.d. | 7.2 | 83.8 | 1015.4 |
| IQR | 8.7 6.5–15.2 | 14.3 12.4–26.7 | n.d. | n.d. | 13.9 1.1–14.6 | 17.7 73.9–91.6 | 11.1 1009.9–1021 |
| Median—cold season | 12.3 | 19.7 | n.d. | n.d. | 1.3 | 90.6 | 1016.1 |
| IQR—cold season | 12.1 7.1–19.3 | 17.3 12.7–30.0 | n.d. | n.d. | 6.8 −2.2–4.6 | 11.0 83.9–94.9 | 15.5 1008–1023.9 |
| Median—warm season | 8.7 | 17.1 | n.d. | n.d. | 14.6 | 76.3 | 1015.1 |
| IQR—warm season | 5.6 6.2–11.7 | 11.4 12.2–23.6 | n.d. | n.d. | 6.3 11.2–17.5 | 15.1 68.3–83.4 | 8.4 1011–1019.1 |

[1] Atm., atmospheric pressure at sea level; [2] Hum., relative humidity; [3] IQR, interquartile range; [4] n.d., no data; [5] $NO_2$, nitrogen dioxide; [6] temp., temperature; [7] $PM_{2.5}$, particulate matter with a diameter of 2.5 μm or less, [8] $PM_{10}$, particulate matter with a diameter of 10 μm or less; [9] $SO_2$, sulfur dioxide.

The meta-analysis of the studied region showed that the IQR increase in $PM_{2.5}$ concentration (OR 1.036, 95% CI 1.016–1.056, *p* < 0.001) was associated with a 3.6% increase in cardiovascular mortality on the day of exposure. The influence of $PM_{2.5}$ persisted on the following days: OR = 1.023 for lag 1 (95% CI 1.003–1.043, *p* = 0.02) and for lag 0–3 OR = 1.038 (95% CI 1.013–1.063, *p* = 0.01). The effect of $PM_{2.5}$ was expressed more in association with ACS-related mortality for lag 0 OR = 1.045 (95% CI 1.012–1.080, *p* = 0.01), and for lag 0–3 OR = 1.040 (95% CI 1.000–1.080, *p* = 0.048).

The IQR increase in concertation of $PM_{10}$ at lag 0 resulted in a 3.4% increase in CVD mortality (1.034, 95% CI 1.015–1.053, *p* < 0.001). The effect was even higher in ACS-related mortality (OR = 1.044, 95% CI 1.010–1.078, *p* = 0.01). The same relation was noted for lag 0–3, with an additional effect on CbVD mortality (OR = 1.099, 95% CI 1.019–1.343, *p* = 0.02). The IQR increase in the $PM_{10}$ concentration was related to a 2.6% increase in CVD mortality (OR = 1.026, 95% CI 1.006–1.047, *p* = 0.01) and a 4.1% increase in ACS-related mortality (OR = 1.041, 95% CI 1.004–1.079, *p* = 0.03). In the meta-analysis of the studied region, no significant impact on cardiovascular mortality was noted for $SO_2$ or $NO_2$. There were no significant differences between the warm and cold season in the meta-analysis of the studied region. Detailed results are shown in Figures 2, 3, S4 and S5.

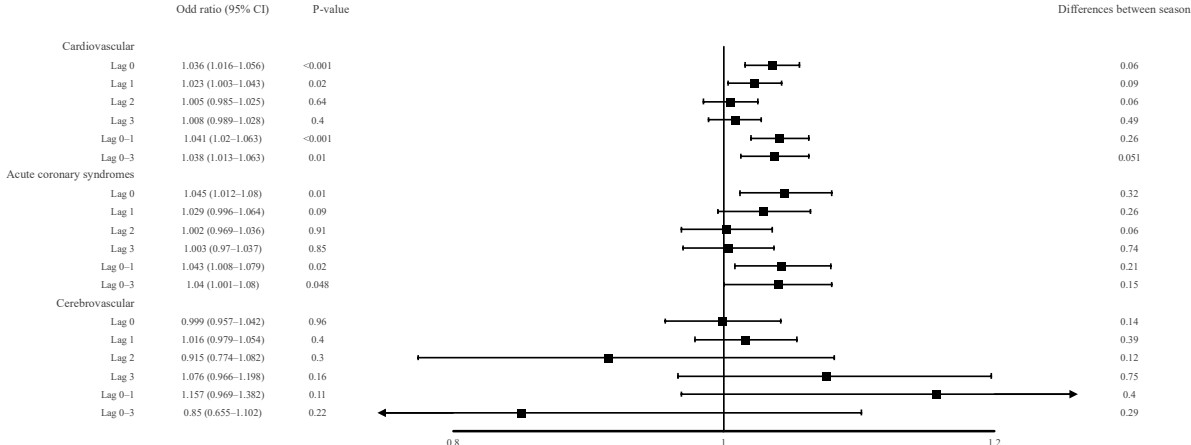

**Figure 2.** Meta-analysis results for the associations between the exposure to short-term particulate matter with a diameter of 2.5 μm or less and cardiovascular-, acute coronary syndrome-, and cerebrovascular-related mortality.

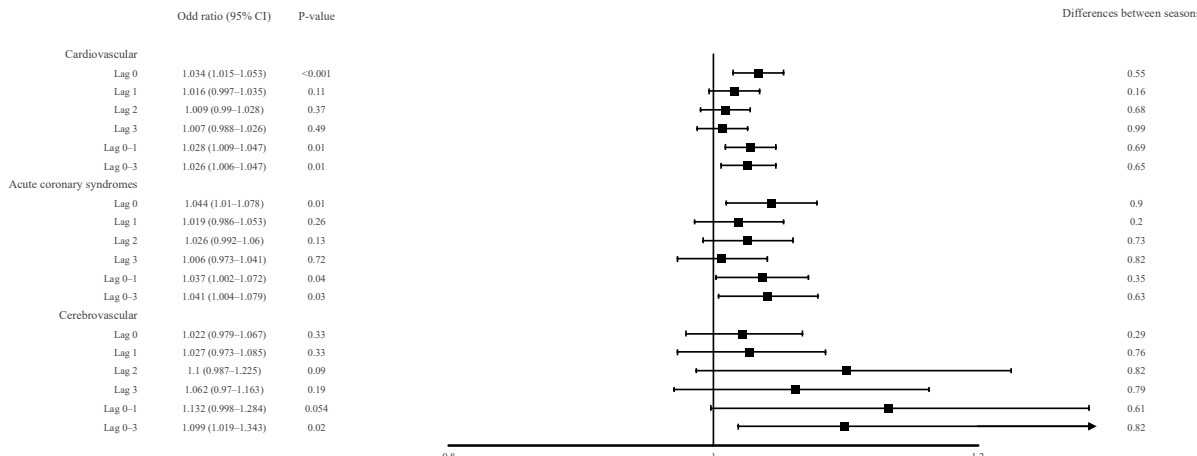

**Figure 3.** Meta-analysis results for the associations between the exposure to short-term particulate matter with a diameter of 10 μm or less and cardiovascular-, acute coronary syndrome-, and cerebrovascular-related mortality.

## 4. Discussion

In 2018, the SDR in Poland was 1218 per 100,000 inhabitants, and it was higher than the mean ratio estimated for the EU, which was 1002 deaths per 100,000 inhabitants [24]. Our analysis showed that the SDRs in all of the of the examined cities were well beyond the estimated rates for the EU, with the highest SDR in Łomża, the city with the greatest contribution of linear sources and the most polluted with $PM_{10}$ (1944.57 deaths per 100,000 inhabitants), thus exceeding the highest SDR in Europe: Bulgaria (1600 deaths per 100,000 inhabitants). The highest rates of ACS and CbVD SDRs were reported in the city with the highest $PM_{2.5}$ concentration median, Białystok. On the other hand, the lowest all-cause and CVD SDRs were also attributed to Białystok. A potential explanation may include the easier access to the physicians and the better long-term medical care of the city residents because tertiary referral hospitals and other highly specialized hospitals concentrate in the region's main city: Białystok.

The major finding of our study is the fact that an impact of air pollution on cardiovascular mortality is also being observed in moderately polluted areas, and this effect is expressed more strongly in the cold season. Although, in our results, we found heterogeneity in the levels of exposure to air pollutants between all of the three cities, each of them presented an association between $PM_{2.5}$ concentrations and CVD mortality on the day of exposure. This adverse health effect was more apparent in ACS and CbVD mor-

tality. Most recently published works report that a $PM_{2.5}$ increase of 10 μg/m$^3$ increases the daily CVD mortality rate by up to 0.55% [25,26]. The concentration–response curves for CVD and all-cause mortality are steeper at lower concentrations of $PM_{2.5}$, especially below 25 μg/m$^3$ [25,27]. Kim et al., in their analysis, reported the highest relative risk of hospitalizations due to CVD at lag 0, with a sharp decline afterwards [25,28]. These facts were partially reflected in our research. Comparing Suwałki—the city with the greatest share of area sources and, concurrently, the least polluted city—to the other cities, the odds ratios of mortality for IQR increased when the $PM_{2.5}$ and $PM_{10}$ concentrations were higher, and the trend was noted regardless of the cause of death up to lag 3. The influence of $PM_{2.5}$ on CbVD and ACS is well documented [29,30]. Ban et al. reported that a 10 μg/m$^3$ increase in $PM_{2.5}$ increases the mortality risk by 0.71%, 1.09% and 0.43% for stroke, ischemic stroke and hemorrhagic stroke, respectively [31]. According to Bourdrel et al., a 10 μg/m$^3$ increment in the $PM_{2.5}$ concentration had an influence on admissions for stroke up to two days after the exposure [32]. Besides the direct influence on ruptures of plaques in arteries, cerebral infarction pathomechanisms may also include the induction of atrial fibrillation, the consequent formation of thrombi, and their release to the bloodstream reaching the brain vessels [33,34].

In our study, significant associations between $PM_{10}$ concentrations and CVD mortality were also noted. Contrary to the ability of $PM_{2.5}$ to penetrate deeply into small alveoli and enter the bloodstream, $PM_{10}$ acts mainly in the upper airways, having a less harmful influence on health [35]. Its increase by 10 μg/m$^3$ increases the daily cardiovascular mortality rate by 0.36% [25]. Some studies claim that $PM_{10}$ has an influence on increased mortality due to CbVD and ischemic heart disease [36]. This thesis is supported by our study, in which the effect of PMs was observed on CVD-, ACS-, and CbVD-related mortality. The detrimental effect was especially expressed in ACS- and CbVD-related mortality.

We noted that CVD mortality associated with $PM_{10}$ was more pronounced in the cold season than the warm season in Suwałki and Białystok—the cities with area–linear pollutant sources that may be linked with the phenomenon called "Polish Smog".

We came to three hypotheses as the possible explanations. The first one states that particulate matter constituents vary by the seasons due to the main sources of air pollution in the analyzed cities. The second takes into consideration the change of the activity patterns of the individuals—more time is spent indoors during the cold season, leading to greater exposure to home heating systems that most often combust coal and wood. The third one suggests that although the PM influence is similar during both seasons, in winter mortality is more visible due to the higher prevalence of infectious diseases.

In our opinion, the explanation of this phenomenon involves the differences in the sources of air pollution in these cities. Surveys revealed that the total ratios of coal- to wood-burning households of Białystok and Suwałki were 0.55 and 3.46, respectively [14,15], which—along with unfavorable weather conditions—lead to creation of the "Polish smog".

Additionally, the greater share of coal burning must have a significant influence on the indoor air pollution in households in Suwałki and Białystok. If we combine this with the aforementioned Peng et al. hypothesis that during cold seasons individuals spend more time indoors, we come to a plausible explanation of this unusual finding. It has been found that the indoor air pollutant concentrations vary during heating and non-heating seasons, sometimes even reaching higher values than those in the ambient air [37,38]. Křůmal et al., in their comparison of the air pollutants emitted from the combustion of coal and biomass in typical boilers used for residential heating in the Czech Republic, reported that the highest PM and polycyclic acrylic acid (PAH) concentrations were linked with coal combustion [38]. Due to the similar socio-economic status of the countries, these results may be more accurately extrapolated onto our study population than studies conducted in Asian or African countries. It was also found that household solid fuel use is associated with an increased stroke risk [39]. Indoor coal combustion processes are known for their contribution to the generation of highly hazardous PAH [40,41].

The important detrimental impact of PAH on CVD was recently the subject of many studies [42]. It has been proven that $PM_{2.5}$-bound PAH concentrations undergo seasonal variations during the year. The most important sources of $PM_{2.5}$-bound PAH are coal and biomass combustion processes during heating seasons, and diesel-related emissions, especially in warm seasons [43]. This fact finds confirmation in the results noted in Łomża, where linear sources prevail and the impact of $PM_{10}$ on ACS-related mortality was higher in the warm season than the cold one. Despite the lower air pollutant concentrations in the warm season, higher traffic, especially cargo and tourist traffic, cause a change in the chemical structure of the air pollutants. The origin of air pollutants has an impact on the chemicals that are bound on the surface of particulate matter [44,45]. When comparing the health effects of various PM sources, diesel exhaust particles (DEPs) have the highest mutagenic activity, generate the most intracellular reactive oxygen species (ROS), and cause altered vascular transcription [46].

Early mechanisms of inhaled pollutants' influence have been mentioned above. The pathophysiology of PAH lies in its ability to activate an oxidative stress-initiated cascade. Numerous studies have proven directly the presence of reactive oxygen species (ROS) themselves and indirectly via biomarkers of oxidative stress and inflammation after exposure to diesel exhaust. ROS reduces the bioavailability of NO—the main factor responsible for vascular tone—leading to endothelial dysfunction. In addition, oxidative stress leads to the proatherogenic oxidation of lipids. They activate endothelial cells, promoting the migration of monocytes that differentiate into macrophages and eventually foam cells. In consequence, chronic exposure leads to the development of plaques and atherosclerosis. Besides this, it has been reported that particulate matter may activate platelets, trigger the release of fibrinogen, tissue factor, and factor VIII whilst inhibiting fibrinolytic capacity. All of the of the listed mechanisms inevitably lead to a higher risk of thrombus formation and potential atherothrombosis [47,48].

In conclusion, as we consider all of the of the above hypotheses to be possible, in our opinion, the aforementioned results provide us a strong indication that not only air pollutant concentrations but also their sources should be considered during the evaluation of the relation between air pollution and mortality. Moreover, while facing the global problem of the COVID-19 pandemic, an air pollution issue might be more actual than ever. Multiple Italian analyses reported that air pollution has an impact on COVID-19 infection, and could act as a co-factor of high-level lethality as chronic exposure to pollutants predisposes an individual to increased susceptibility to infectious agents [49,50].

It is important to remember that the harmful effects of air pollution on human health are not the result of a single pollutant, but rather of a mixture of many toxic substances. Any results showing the detrimental effects of air pollution should exert pressure to implement systemic changes and to increase ecological awareness among the population, which combined together could significantly improve air quality in the long term. For now, precautionary and preventive measures should be enhanced, especially for people from high-risk groups.

## 5. Conclusions

The short-term impact of PMs on cardiovascular mortality are also observed in moderately polluted areas. This adverse health effect was more apparent in CbVD and ACS mortality, and in the cold season. The differences in the effect size and seasonality may depend on the source of air pollution, and further research focused on the adverse health effect of "Polish smog" is needed. Our findings should encourage policymakers to implement strategies focused on the reduction of the exposure to area-source air pollution in moderately polluted areas.

**Supplementary Materials:** The following are available online at https://www.mdpi.com/article/10.3390/atmos12101270/s1. Table S1: Crude cause-specific cardiovascular mortality in the studied cities, 2008–2017. Table S2: Time-stratified case-crossover model for the cities included in the meta-analysis. The odds ratio of cardiovascular mortality with interquartile-range increase in the exposure to air pollutants. Table S3: Time-stratified case-crossover model for the cities included in the meta-analysis. The odds ratio of acute coronary syndromes with the interquartile-range increase in the exposure to air pollutants. Table S4: Time-stratified case-crossover model for the cities included in the meta-analysis. The odds ratio of cerebrovascular mortality with the interquartile-range increase in the exposure to air pollutants, Figure S1: Panel chart. Changes in the concentrations of air pollutants and the temperature in Białystok for the analyzed period (the red line represents changes in the quartile of the year). Figure S2: Panel chart. Changes in the concentrations of air pollutants and the temperature in Łomża for the analyzed period (the red line represents changes in the quartile of the year. Figure S3: Panel chart. Changes in the concentrations of air pollutants and the temperature in Suwałki for the analyzed period (the red line represents changes in the quartile of the year). Figure S4: Meta-analysis results for associations between exposure to short-term nitrogen dioxide and cardiovascular-, acute coronary syndromes-, and cerebrovascular-related mortality. Figure S5: Meta-analysis results for associations between short-term exposure to sulfur dioxide and cardiovascular-, acute coronary syndromes-, and cerebrovascular-related mortality.

**Author Contributions:** Conceptualization, Ł.K., A.K. and H.B.-G.; methodology, Ł.K.; formal analysis, Ł.K., E.J.D. and S.D.; investigation, A.K. and E.J.D.; data curation, Ł.K.; writing—original draft preparation, Ł.K., A.K. and E.J.D.; writing—review and editing, Ł.K., A.K. and E.J.D.; visualization, Ł.K.; supervision, S.D. and H.B.-G. All authors have read and agreed to the published version of the manuscript.

**Funding:** This research received no external funding.

**Institutional Review Board Statement:** The study was conducted according to the guidelines of the Declaration of Helsinki, and approved by the Ethics Committee of the Medical University of Białystok (Approval No. R-1-002/18/2019).

**Informed Consent Statement:** Informed consent was obtained from all of the subjects involved in the study.

**Data Availability Statement:** Derived data supporting the findings of this study are available from the corresponding author on request.

**Conflicts of Interest:** The authors declare no conflict of interest.

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
