# Peer review of "Short-Term Effects of “Polish Smog” on Cardiovascular Mortality in the Green Lungs of Poland: A Case-Crossover Study with 4,500,000 Person-Years (PL-PARTICLES Study)"

_atmosphere, doi:10.3390/atmos12101270_

Round 1

Reviewer 1 Report

This is a well design and well written manuscript to demonstrate the health effect of air pollution. I have just only one minor suggestion for authors should show the definition of the "short-term" cardiovascular mortality effect.

Author Response

Reviewer 1

This is a well design and well written manuscript to demonstrate the health effect of air pollution. I have just only one minor suggestion for authors should show the definition of the "short-term" cardiovascular mortality effect.

We are very grateful for the time and effort invested in reviewing our manuscript as well as the kind words. We have updated the methods section so that it clearly states the definition of "short-term" and it now states as follows:

“We estimated the effects of air pollutants at single-day lag (from lag 0 to lag 3) and at multi-day lag (lag 0-1 and lag 0-3), which defines the short-term period as a maximum of 3 days. For example, lag 0 and lag 0-1 correspond to the concentration of pollutants on the death day and moving average for the death day and one previous day, respectively.”

Reviewer 2 Report

Thank you for the opportunity to revise this interesting manuscript, which deals with a pivotal environmental and public health issue, i.e. air pollution and risk of mortality.

I have found the study design appropriate, and the methods clearly presented. The discussion is thorough and the conclusions are supported by the findings.

Overall I really enjoyed reading the manuscript, and I believe it will be of interest to the readers of the Journal, as it falls within its aims and scope.

I have only a few minor suggestions the authors could consider, which are listed below.

In the methods section, the author could report the formula used to standardize the mortality ratio. 

In the results section, the authors should avoid making any discussion or consideration, more appropriate in the discussion section.

In the introduction or Discussion section, the authors could consider adding some references related to the role of air pollutants and the COVID-19 mortality. In particular, different hypotheses have been studied in order to understand why the SARS-CoV-2 virus has initially spread in some specific geographical areas. We know that Italy has been hit as the first country in the western area after the virus has appeared. One of the main hypotheses was related to the environmental and air pollution of specific areas (i.e., the Po Valley), in particular referring to the chronic exposure of the population to air pollutants. This condition is known to increase the risk of different health problems, such as infectious diseases and cardiovascular diseases.

I believe this aspect could be of interest and could strengthen the research question the authors has dealt with. I could suggest some recent papers the authors could consider: I) DOI: 10.1016/j.envres.2020.110459; II) https://doi.org/10.3390/su12125064; III) https://doi.org/10.1016/j.envpol.2020.114465; IV)   https://doi.org/10.1136/bmjopen-2020-039338. 

I hope the authors will follow the suggestions, as I believe the manuscript will be well-received by the scientific community.
